# Automated markerless pose estimation in freely moving macaques with OpenMonkeyStudio

Praneet C. Bala[1], Benjamin R. Eisenreich[2], Seng Bum Michael Yoo[2], Benjamin Y. Hayden [2,3,4,5✉], Hyun Soo Park[1,5] & Jan Zimmermann[2,3,4,5]

The rhesus macaque is an important model species in several branches of science, including neuroscience, psychology, ethology, and medicine. The utility of the macaque model would be greatly enhanced by the ability to precisely measure behavior in freely moving conditions. Existing approaches do not provide sufficient tracking. Here, we describe OpenMonkeyStudio, a deep learning-based markerless motion capture system for estimating 3D pose in freely moving macaques in large unconstrained environments. Our system makes use of 62 machine vision cameras that encircle an open 2.45 m × 2.45 m × 2.75 m enclosure. The resulting multiview image streams allow for data augmentation via 3D-reconstruction of annotated images to train a robust view-invariant deep neural network. This view invariance represents an important advance over previous markerless 2D tracking approaches, and allows fully automatic pose inference on unconstrained natural motion. We show that OpenMonkeyStudio can be used to accurately recognize actions and track social interactions.

[1] Department of Computer Science and Engineering, University of Minnesota, Minneapolis, MN 55455, USA. [2] Department of Neuroscience, University of Minnesota, Minneapolis, MN 55455, USA. [3] Center for Magnetic Resonance Research, University of Minnesota, Minneapolis, MN 55455, USA. [4] Center for Neuroengineering, University of Minnesota, Minneapolis, MN 55455, USA. [5]These authors contributed equally: Benjamin Y. Hayden, Hyun Soo Park, Jan Zimmermann. ✉email: benhayden@gmail.com

Rhesus macaques are one of the most important model organisms in the life sciences, e.g., refs. [1–3]. They are invaluable stand-ins for humans in neuroscience and psychology. They are a standard comparison species in comparative psychology. They are a well studied group in ethology, behavioral ecology, and animal psychology. They are crucial disease models for infection, stroke, heart disease, AIDS, and several others. In all of these domains of research, characterization of macaque behavior provides an indispensable source of data for hypothesis testing. Macaques evolved to move gracefully through large three-dimensional spaces (3D) using four limbs coordinated with head, body, and tail movement. The details of this 3D movement provide a rich stream of information about the macaque's behavioral state, allowing us to draw inferences about the interaction between the animal and its world[4–9]. We typically measure only a fraction of available information about body movement generated by our research subjects. For example, joystick, button press, and gazetracking measure a very limited range of motion from a single modality. One could potentially incorporate more such measurement devices, but there are practical limits in training and use. More broadly, it is possible to divide movement into actions that take account of the entire body by delineating an ethogram, which expressly characterizes and interprets full body positions and actions (see, for example, Sade, 1973). However, ethograms can generally only be done by highly trained human observers, are labor intensive, costly, imprecise, and susceptible to human judgment errors[10]. These limitations greatly constrain the types of science that can be done and therefore the potential value of that research.

For these reasons, the automated measurements of 3D macaque pose is an important goal[11,12]. Pose, here, refers to a precise description of the position of all major body parts (landmarks) in relation to each other and to the physical environment. Pose estimation can currently be done with a high degree of accuracy by commercial marker-based motion capture systems (e.g., Vicon, OptiTrack, and PhaseSpace). Macaques, however, are particularly ill-suited for these marker-based systems. Their long, dense, and fast-growing fur makes most machine-detectable markers difficult to attach and creates a great deal of occlusion. Their highly flexible skin makes markers shift position relative to bone structure during vigorous movement, which is common. Their agile hands and natural curiosity make them likely to remove most markers. They often show discomfort, and consequently unnatural movement regimes, with jackets and bodysuits.

Markerless motion capture offers the best possibility for a widely usable tracking system for macaques. Recent success in deep learning-based 2D human pose estimation from RGB images[13–15] opens a new opportunity for animal markerless motion capture. However, due to the millions of trainable parameters in deep neural networks, pose estimation requires a large quantity of training data. Transfer learning is a promising approach to alleviating the data requirement by providing a small set of new domain (scene)-specific annotations for a testing scene. The result of transfer learning, generalization within the testing scene can be remarkable. For instance, DeepLabCut leverages a pretrained deep learning model (based on ImageNet) to accurately localize body landmarks. Methods that make use of transfer learning work for various organisms like flies, worms, and mice by learning from a larger number of images collected from a single view[16–19]. Two extremely promising extensions towards view-invariant pose estimation in animal models have recently been demonstrated in cheetahs[17] and in flies by integrating active learning[20].

However, macaques present several problems that make current best markerless motion capture unworkable. First, they have a much greater range of possible body movements than other model organisms. Most notably, each body joint has multiple degrees of freedom, which generates a large number of distinctive poses associated with common activities such as bipedal/quadrupedal locomotion, grooming, and social interactions in even modestly sized environments. Second, they interact with the world in a fundamentally three-dimensional way, and so they must be tracked in 3D. Existing 2D motion tracking learned from the visual data recorded by a single-view camera can only produce a view-dependent 2D representation. Thus, application to novel vantage points introduces substantial performance degradation.

These visual characteristics of macaques are analogous to those of humans; the tracking problem in humans has been effectively solved by fully supervised learning with a large database of annotated images (e.g., 2.5 million annotated images[21–23]). Critically, this large dataset allows strong generalization even in novel testing scenes (e.g., different lighting, poses, views, identities, and appearance). However, such a database does not exist for macaques and would be prohibitively expensive to generate. Specifically, we estimate that to generate a macaque database similar to the one used for humans, it would take hundreds of thousands of distinct annotated images for each joint, and cost roughly $10 M. Thus, successful tracking requires a larger pipeline that includes generating a large annotated pose database. This annotation problem simply cannot be overcome by better AI, and requires a qualitatively new approach.

Here, we present a description of OpenMonkeyStudio, a deep learning-based markerless motion capture system for rhesus macaques (Fig. 1) based on fully supervised learning. It solves the annotation problem through innovations in image acquisition, annotation and label generation, and augmentation through multiview 3D reconstruction. It then implements pose estimation using a deep neural network. Our system uses 62 cameras, which provides multiview image streams that can augment annotated data to a remarkable extent by leveraging 3D multiview geometry. However, while this large number of cameras is critical for training the pose detector, the resulting model can be used in other systems (for example by other laboratories) with fewer cameras without training. Our system generalizes readily across subjects and can simultaneously track two individuals. It is complemented by the OpenMonkeyPose dataset, a large database of annotated images (195,228 images) which we will make publicly available.

## Results

**Markerless motion capture for macaques using OpenMonkeyStudio.** We developed a multiview markerless motion capture system called OpenMonkeyStudio that reconstructs a full set of three-dimensional (3D) body landmarks (13 joints) in freely moving macaques (Fig. 2) without manual intervention. The system is composed of 62 synchronized high definition cameras that encircle a large open space (2.4 × 2.45 × 2.75 m) and observe a macaque's full body motion from all possible vantage points. For each image, a pose detector made of a deep neural network predicts a set of 2D locations of body landmarks that are triangulated to form the 3D pose given the camera calibration parameters (focal length, lens distortion, rotation, and translation) as shown in Fig. 1.

We built the pose detector using a Convolutional Pose Machine (CPM[24],), which learns the appearance of body landmarks (e.g., head, neck, and elbow) and their spatial relationship from a large number of annotated pose instances. Note that our framework is agnostic to the design of the underlying network, and therefore, other landmark detectors such

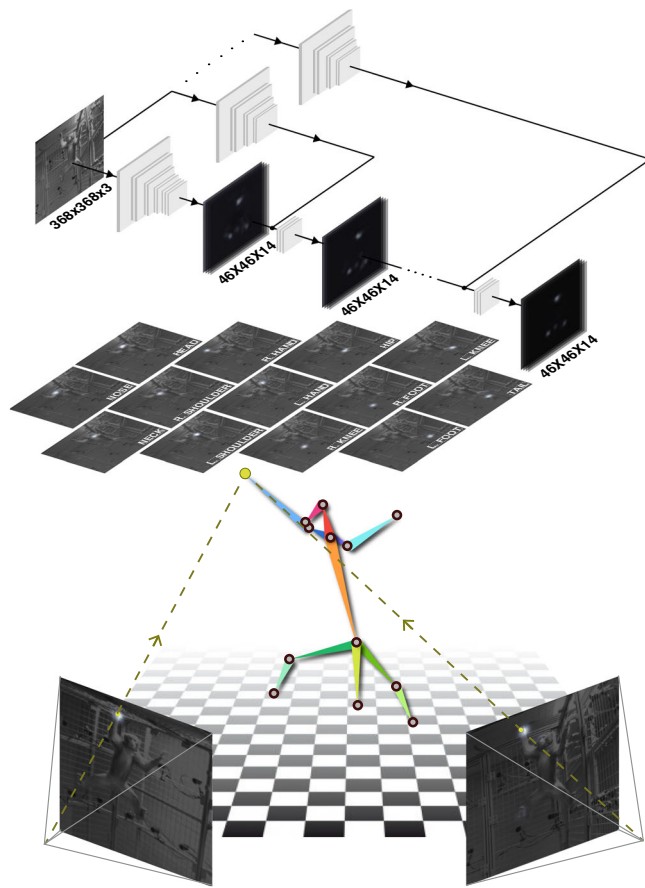

**Fig. 1 Pose detection architecture.** A multi-stage convolutional pose machine[24] is used to detect body landmarks of macaque from an image. It takes as an input a 368 × 368 × 3 image (368 × 368 resolution with three color channels) and outputs 46 × 46 × 14 response maps (13 landmarks and one background) where the location of maximum response corresponds to the landmark location. The detected land-marks from multiview images are triangulated in 3D given the camera calibration[41]. To train the generalizable view-invariant pose detector, multiview geometry is used to substantially augment the data via 3D reconstruction, which allows learning a view-invariant pose detector.

as Stacked Hourglass[14] or DeeperCut[15] (also used in DeepLab-Cut[16]) can be complementary to the CPM.

**Quantitative evaluation of OpenMonkeyStudio**. Here we describe validations of the OpenMonkeyStudio system including its accuracy, effectiveness, and precision.

We evaluated the accuracy of head pose reconstruction by comparing to the best available marker-based video motion capture system (OptiTrack, NaturalPoint, Inc, Corvallis, OR). We chose the head because it is the only location where a marker can be reliably attached without disturbing free movement in macaques. Figure 3 illustrates the head trajectory in three dimensions as measured by both methods over 13 min measured at 30 Hz. This illustrative sequence includes jumping and climbing (insets). Assuming that the OptiTrack system represents the ground truth (i.e., 0 error), the reconstruction by Open-MonkeyStudio has a median error of 6.76 cm, a mean error of 7.14 cm and a standard deviation: 2.34 cm. Note that there is a spatial bias due to marker attachment (roughly 5 cm above the head), which inflates the presented error estimate. In addition to this systematic offset, our training data did not include

instrumented (implant carrying) animals, which could further increase variability in the location estimate. Note that these presumptive ground truth data from OptiTrack include obvious and frequent excursion errors as shown in Fig. 3. This is caused by marker confusion and occlusion, which requires additional manual post-processing to remove. Additionally, infrared inter-ference is a significant hurdle in a large enclosure such as ours. By contrast, OpenMonkeyStudio leverages visual semantics (appear-ance and spatial pose configuration) from images that can automatically associate the landmarks across time. That in turn makes our system more robust to confusion/occlusion errors, a common failure point in any motion capture system. It can even predict the occluded landmarks based on the learned spatial configuration, e.g., knowing shoulder and elbow joints is highly indicative of the occluded hand's location. We construct occlusion agnostic training data to handle occlusion, i.e., we use projections of 3D reconstructed pose for labeling data, which enforces the network to predict the locations of occluded joints.

We evaluated the effectiveness of multiview augmentation that is used to generate the training data, i.e., how many cameras are needed for training. We measured the relative accuracy of tracking as a function of the number of cameras, each of which supplements view augmentation ($m = 1, 2, 4, 8, 16, 32$, and 48 cameras) compared to the model generated from the full system ($m = 62$ cameras). For each pose detector trained by the augmentation with a factor of $m$, we reconstructed the 3D pose using $n = 62$ cameras for a new testing sequence on different macaques. (Here, $m$ and $n$ denote the number of cameras used for training and testing, respectively). Among 62 cameras, we select the views for augmentation uniformly over camera placement. We compared the reconstructed poses with the pseudo ground truth reconstructed by the full model. Note that $m = 1$ camera and $m = 2$ cameras are equivalent to the single-view approach[16] and stereo view approach[25]. These are the special instances of OpenMonkeyStudio with limited view augmentation.

Figure 4 illustrates the relative accuracy measured by the percentage of correct reconstruction for each landmark, i.e., how many testing instances are correctly reconstructed given the error tolerance (10 cm). For visually distinctive and relatively rigid landmarks, such as nose, neck and head, relatively accurate reconstruction can be achieved by a small number of augmenta-tions, e.g., training with a single-view camera can produce ~65% of correct reconstruction. Note that even for these ostensibly "easy" landmarks, augmentation still provides substantial bene-fits. However, for the limb landmarks that have higher degrees of freedom (hands, knees, and feet) and are frequently self-occluded, their reconstructions are in particular vulnerable to a viewpoint variation because the appearance of such landmarks varies significantly across views. This leads to considerable performance degradation (for example, 12% of correct reconstruction for the single view). Overall performance (black line) is increased from 34% ($m = 1$) to 76% ($m = 48$), justifying the multiview augmentation.

We next evaluated the inference precision, i.e., how many cameras are needed for inference (testing data) to produce comparable reconstruction with 62 cameras. We measured the precision of 3D reconstruction of the full model ($m = 62$) while varying the number of views ($n = 2, 4, 8, 16, 32$, and 48) for a testing sequence with different macaques. $n = 1$ is impossible as 3D reconstruction requires at least two views. The error measure (the percentage of correct reconstruction for each landmark) is identical to the relative accuracy analysis above (Fig. 4). The inference precision quickly approaches 80% average performance of 13 landmarks with as few as eight cameras. However, as with view augmentation for training, hands, knees, and feet require more cameras ($n = 32$). In other words, to fully capture the

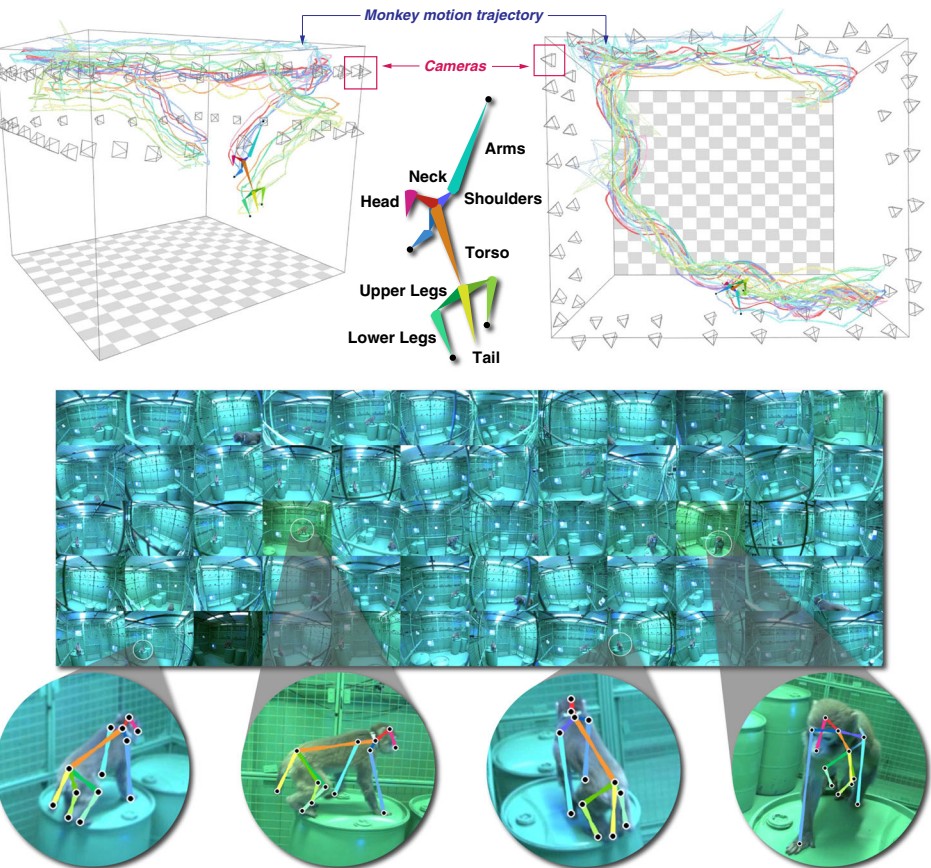

**Fig. 2 Representative tracking results.** A markerless motion capture system called OpenMonkeyStudio is designed to reconstruct 13 body landmarks in 3D. The system with 62 cameras that encircle a large open space synchronously captures a macaque's movement from diverse vantage points. Colored traces correspond to joint labels. The multiview images and four arbitrary cropped images superimposed with the projection of the reconstruction are shown.

position of the extremities, there is a strong benefit to using dozens of cameras.

Although the trained pose detector was designed to be view-invariant, in practice, it can still be view-variant. That is, the inference precision can still depend on viewpoint. We characterize the view-dependency in Fig. 5. Specifically, we illustrate the accuracy of the 2D pose inference by comparing decimated reconstructions to the presumed ground truth of full reconstruction. The views are organized with respect to the macaque's facing direction (detected automatically using the detected head pose). Specifically, the relative camera angle is negative if the cameras are located on the left side, and positive otherwise. We find that for the landmarks that are visible in most views (such as head and pelvis), the 2D localization is highly view-invariant, meaning that there is uniform accuracy across views (less than 2 pixel error). However, for the hands and feet, which are frequently occluded (often, by the torso), localization is often highly view-variant. For example, right-hand side views are typically less suitable to localize the left hand. This view-variance can be alleviated by leveraging multiple views. Nonetheless, to ensure the minimal view-variance in 3D reconstruction, the cameras need to be distributed uniformly across the enclosed space.

**Automated identification of semantically meaningful actions.** The central goal of tracking, of course, is to identify actions[10]. The ability to infer actions from our data, is therefore a crucial measure of the effectiveness of our system. We call this expressibility. We therefore next assessed expressibility for our 3D representations and, for comparison, our 2D representations.

For processing 3D representations, we transformed each one-frame pose into a canonical coordinate system. Specifically, we defined the neck as the origin and the y-axis as the gravity direction. The $z$-axis is defined as aligned with the spine (specifically, the axis connecting the neck and hip), and the representation is then normalized so that the length of the spine is one. This coordinate transform standardizes the location, orientation, and size of all macaques. We then vectorize the 3D transformed landmark coordinates to form the 3D representation. That is, $[x_{head} y_{head} z_{head} \cdots x_{tail} y_{tail} z_{tail}]^T \in \mathbb{R}^{36}$ (note that the neck location is not included as it corresponds to the origin.).

In Fig. 6a, we visualize the clusters of the 3D representations of macaque movements in an exemplary 30 min sequence (that is, 54,000 frames). We used Uniform Manifold Approximation and Projection (UMAP) for dimensionality reduction[26]. This process results in coherent clusters. Visual inspection of these clusters in turn demonstrates that they are highly correlated with the semantic actions such as sitting, standing, climbing, and climbing upside down. We further use these clusters to classify actions in a new testing sequence (13 min) using a k nearest neighbor search. This allows identifying the transitions among these actions as shown in Fig. 6b. The transitions are physically sensible, e.g., from walking on the floor to climbing upside down, a macaque needs transitional actions of walking → standing →climbing → climbing upside down.

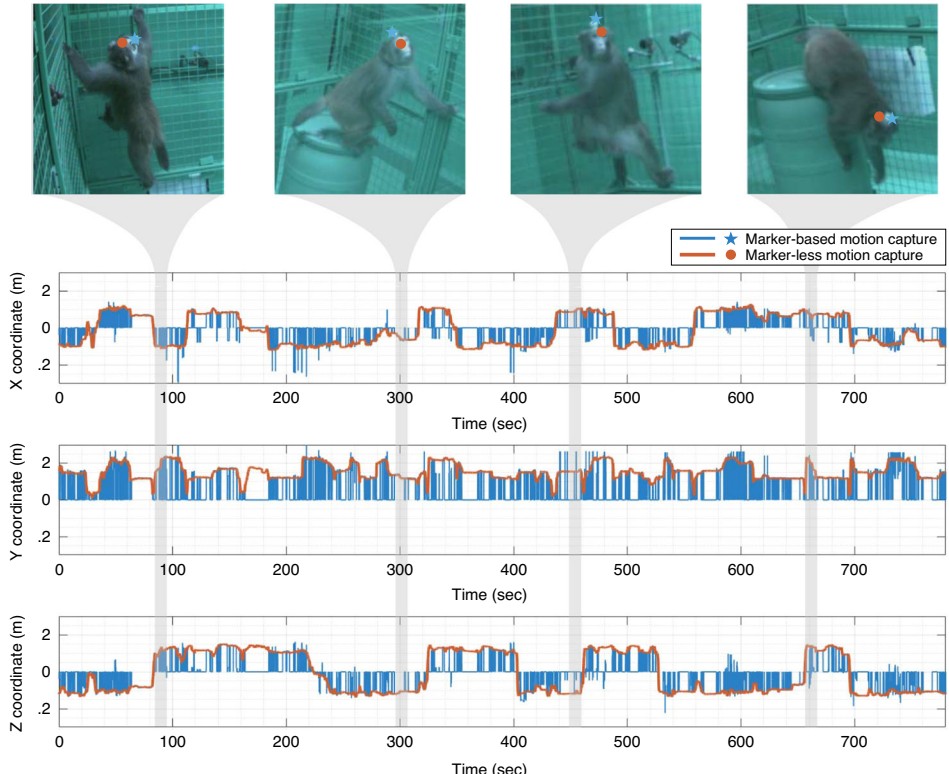

**Fig. 3 Pose tracking validation.** The head location reconstructed by OpenMonkeyStudio is compared with a marker-based motion capture system (OptiTrack) over time. The marker-based system produces noisy measurements due to the marker confusion, which requires an additional manual refinement. The median error is 6.76 cm. The images overlaid with the projection of the 3D reconstruction show visual validity.

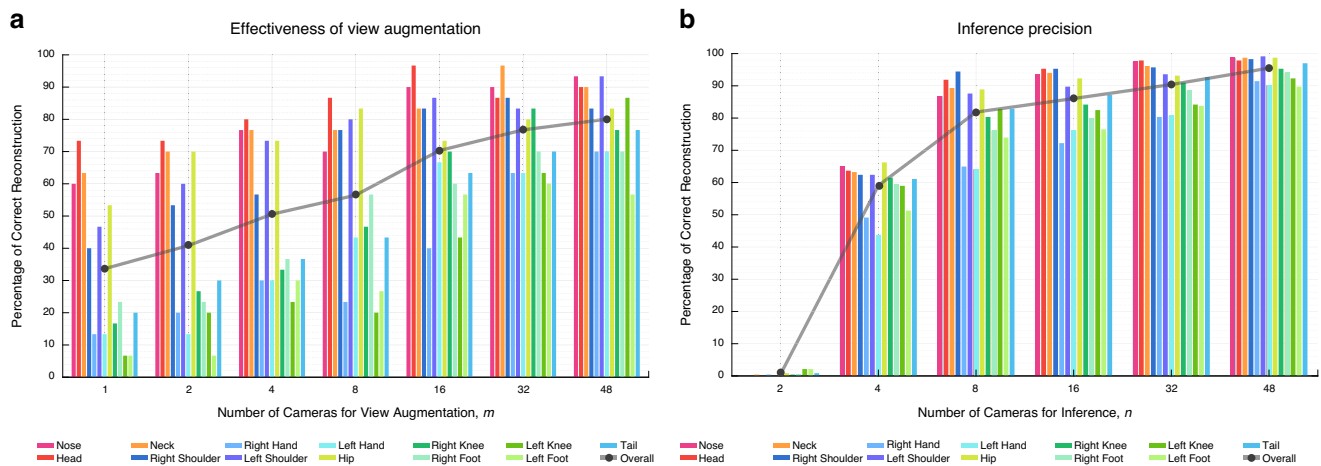

**Fig. 4 Augmentation and inference performance. a** View augmentation improves the accuracy of reconstruction. A subset of cameras are used for view augmentation and the relative accuracy is measured by comparing to the full model that is trained by 62 camera augmentation ($m = 62$). While the landmarks that are relatively rigid shape such as nose, head, neck, and hip can produce accurate reconstruction with small augmentation, the limb landmarks such as hands, knees, and feet require greater augmentation. The overall accuracy is improved from 34% ($m = 1$) to 76% ($m = 48$), which justifies the multiview augmentation. **b** Once the detection model is trained with the full view augmentation ($m = 62$), a subset of cameras can be used to achieve comparable performance. The relative accuracy is measured by comparing to $n = 62$. For instance, eight cameras can achieve 80% overall performance. However, the limbs with high degrees of freedom such as hands, knees, and feet require more cameras to reach comparable levels.

For comparison, we next performed the same analyses on 2D representations. To process the 2D representations, we first vectorized the set of landmarks in each frame, thus, $[x_{\text{head}} y_{\text{head}} \dots x_{\text{tail}} y_{\text{tail}}]^{\text{T}} \in \mathbb{R}^{26}$ (The number 26 comes from the fact that we have 13 joints in each of two dimensions). In contrast, the 2D representation is affected by viewpoint where the clusters are not distributed in a semantically meaningful way as shown in Fig. 6c.

**Social interaction**. Macaques, like all simian primates, are highly social animals, and their social interactions are a major determinant of their reproductive success, as well as a means of

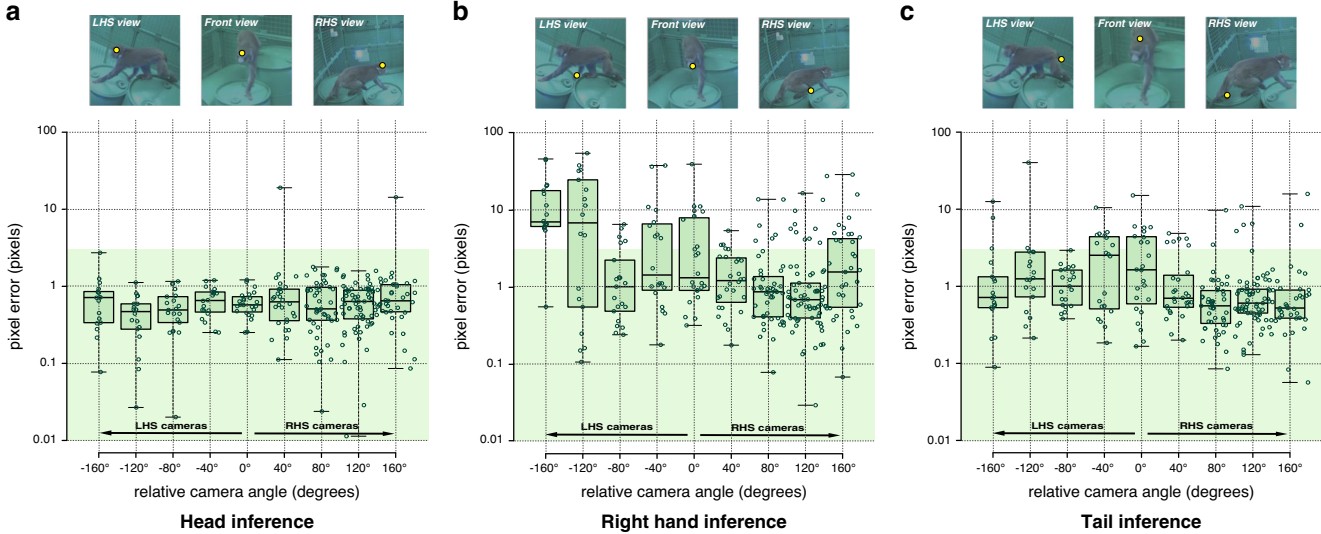

**Fig. 5 View-dependent accuracy.** The inference precision of landmark detection is view dependent. For the head (**a**) that is visible from most views, the precision is nearly uniform across views under the valid inference range (two pixel error, $n = 344$ images). In contrast, the right hand (**b**) is often occluded by the torso when seen from the cameras on the left-hand side of the macaque. The tail (**c**) corresponds more to the head inference with a slight degradation when the macaque is seen from head on. This results in non-uniform precision, i.e., the inference from the views on the right-hand side is more reliable than the other side. Boxplots represent median and 25th and 75th percentile respectively while whiskers extend to extrema.

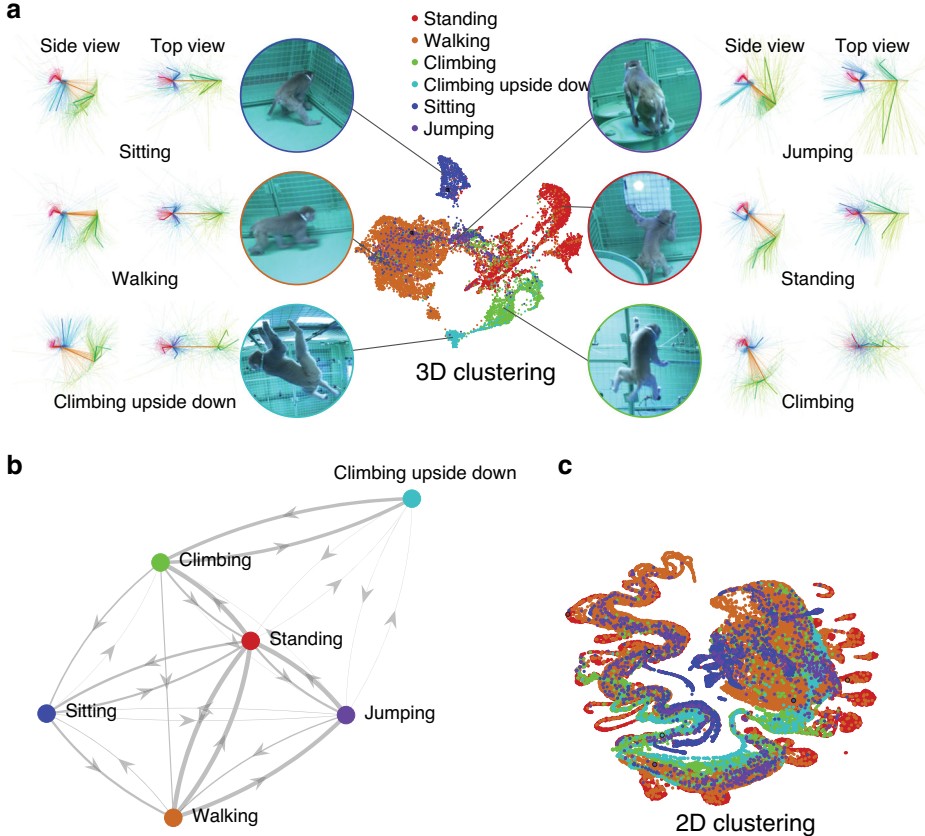

**Fig. 6 Semantic action detection.** We use the 3D pose representation to recognize semantic actions (standing, walking, climbing, climbing supine, sitting, and jumping). **a** The poses are clustered by using UMAP. Each cluster that is represented by 3D poses (side and top views) is highly correlated with the semantic actions. **b** With the clusters, we recognize actions in a new testing sequence using the k nearest neighbor search and visualize the transitions among the semantic actions. **c** In contrast, the 2D representation provides the clusters that are driven by the pose and viewpoint. For instance, while the 3D representation of walking is one continuous cluster, the 2D representation is broken apart into discrete groupings of repeated poses at different spatial locations.

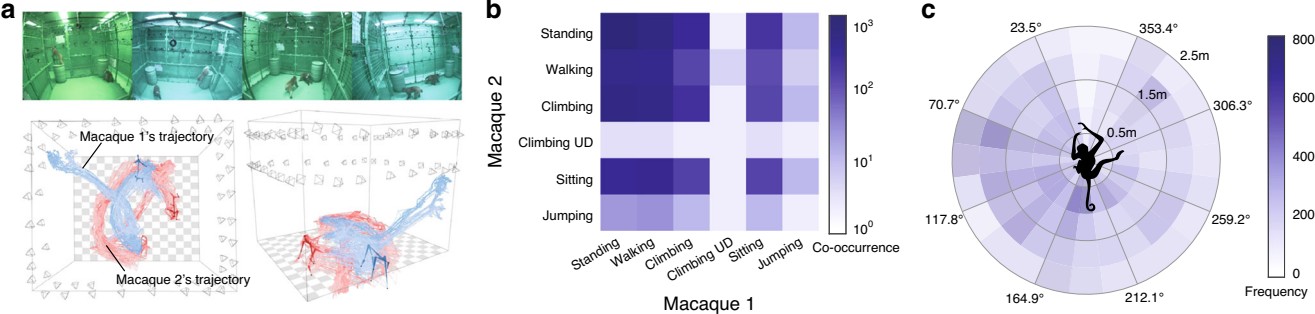

**Fig. 7 Social interaction tracking. a** OpenMonkeyStudio extends to tracking social interactions in non-human primates. Here we demonstrate the feasibility of tracking two rhesus macaque while they individually move inside the enclosure, crossing paths fully. Colors indicate two individuals. Top frames depict the scene of two individuals in the cage during different timepoints. **b** Co-occurrence of actions in social macaques. We used 3D poses to classify their actions to illustrate the co-occurrence of actions of two macaques in log scale. **c** Proxemics characterizes the social space, e.g., how the location of a macaque is distributed with respect to the other. We transformed the 3D coordinate of the second macaque to the first macaque's body centric coordinate system, i.e., 0° represents the first macaque's facing direction. We use the polar histogram of the transformed coordinate to visualize the proxemics of macaques.

communication[27–30]. OpenMonkeyStudio offers the ability to measure multiple macaque poses jointly. To show its capability, we first generated a dataset using two macaques placed together in our large cage system. The two subjects were familiar with each other and exhibited clear behavioral tolerance in their home caging. These two macaques freely navigated in the open environment while interacting with each other. Figure 7a illustrates the 3D reconstruction of their poses over time. It is important to note that extraction of the animal's field of view is highly dependent on the number of cameras available thus further justifying our high camera count. The closer animals are to each other, the more important are unique views that are able to separate the individuals. Figure 7b demonstrates that the same tools and approaches used for single macaque pose representation readily extend to social interactions: the co-occurrences of actions of two macaques. This approach illustrated in Fig. 7c allows us to compute proxemics[31] for two macaques interacting. That is, it lets us compute the average position of each macaque relative to the other, including information about orientation. The relative frequency of a conspecific in a given position relative to a focal subject defines the proxemic relationship of the pair. This information in turn can be used to estimate average distance between subjects, average angle between them, and the interaction of those two terms. Proxemics can be plotted using a radial plot (Fig. 7c). This plot illustrates the proxemics of two macaque subjects. We use the polar histogram of the transformed coordinate to indicate frequency of co-occurrence of the two subjects (color, z-dimension of plot) as a function of their relative distance (r dimension) and angle (theta dimension).

**OpenMonkeyPose dataset**. We have argued that the critical barrier to tracking is overcoming the annotation problem (see above). We have done so here by leveraging a large-scale multi-view dataset of macaques to build the pose detector for Open-MonkeyStudio. Our dataset consists of 195,228 image instances that can densely span a large variation of poses and positions seen from 62 views (Fig. 8). The dataset includes diverse configurations of the open unconstrained environment, and also involves inanimate objects (barrels, ropes, feeding stations). It also involves multiple camera configurations and types, it involves two background colors (beige and chroma-key green), and four macaque subjects varying in size and age (5.5–12 kg). The dataset, trained detection model, and training code is available at https://github.com/OpenMonkeyStudio.

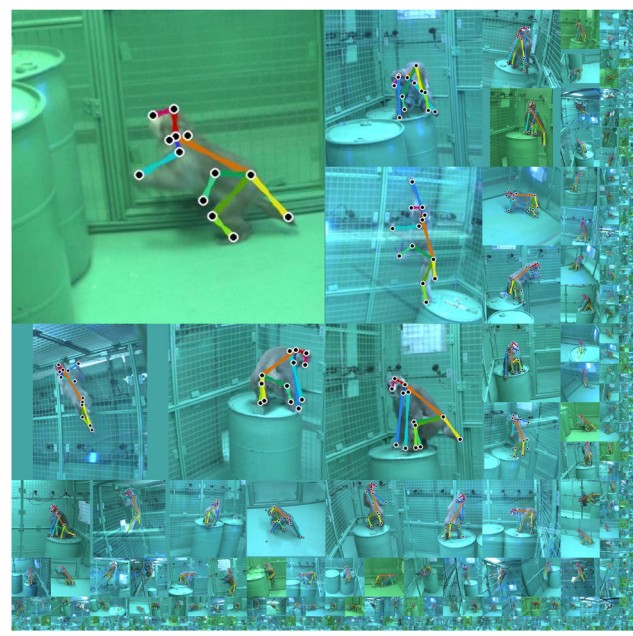

**Fig. 8 Dataset.** We will make OpenMonkeyPose dataset, the trained detection model, and the training code publicly available. The dataset includes 195,228 annotated pose instances associated with diverse activities.

## Discussion

Here, we present OpenMonkeyStudio, a method for tracking the pose of rhesus macaques without the use of markers. Our method makes use of 62 precisely arranged high-resolution video cameras, a deep learning pose detector, and 3D reconstruction to fit rich behavior. Our system can track monkey pose (13 joints) with high spatial and temporal resolution for several hours, which has been impossible with a marker-based motion capture. It can track two interacting monkeys and can consistently identify individuals over time. The ability to track positions of macaques is important because of their central role in biomedical research, as well as their importance in psychology, and ethology. Recent years have witnessed the development of widely used markerless tracking systems in many species, including flies, worms, mice and rats, and humans[16,20,32,33]. Such systems are typically not designed

with the specific problems of monkey pose estimation in mind. Relative to other more readily trackable species, monkeys have largely homogeneous unsegmented appearances (due to their thick continuous and mostly single colored fur covering), have much richer pose repertoires, and have much richer positional repertoires. (Consider, for example, that a mouse rarely moves to a sitting position, much less flips upside down). Although our system takes multiple approaches to solve these problems, the major innovation concerns overcoming the annotation problem. That is, the major barrier to successful pose tracking in macaques is the lack of a sufficiently large reliably annotated training set, rather than the lack of an algorithm that can estimate pose given that set. A similar approach has recently been successfully applied in the tracking of flies constrained by tethers[20] on which our work naturally builds and extends to generalized free movement.

The annotation problem is deceptively complex. We estimate that to generate a dataset of quality equivalent to that used in human studies would cost a few million dollars—several orders of magnitude more than our system costs. We get around this problem using several innovations: (1) a design of a dense multi-camera system that can link multiview images through a common 3D macaque's pose, (2) an algorithm that strategically selects maximally informative frames, thus allowing us to allocate manual annotation efforts much more efficiently, and (3) multi-view augmentation, or increasing the effective size of our dataset by leveraging 3D reconstruction. We further augment these with a standard additional step, affine transformation. These steps, combined with professional annotation of a subset of data, allow for the creation of a dataset called OpenMonkeyPose sufficient for machine learning, which we will make publicly available at the time of publication. Although our system is designed for a single cage environment, it can readily be extended to other environment shapes and sizes. We demonstrate that with the trained pose detector, it is possible to reduce the density of cameras for other environments, e.g., for $2.45 \times 2.45 \times 2.75$ m space, eight cameras produce 80% performance compared to 62 cameras ($1280 \times 1024$). For a larger space, higher spatial resolution is needed, which can effectively produce a similar size of region of interest that contains a macaque.

OpenMonkeyStudio is a blueprint for tracking pose in monkeys. As such, engineering expertise is essential in repli-cating our approach. Our analysis provides strong evidence that generalization to other laboratory environments is possible; however, we are currently not providing a plug and play solu-tion. Replication of the system we have developed requires roughly the following technical steps. First a larger scale enclosure (preferably homogeneously painted) and monkey transfer mechanism has to be built. A networked large-scale camera system has to be deployed and tested for temporal precision. Calibration routines have to be tested and cameras placed, focused and positioned to minimize calibration errors. Next, initial image acquisitions with monkeys have to be per-formed. Lastly, image segmentation routines provided in our tools have to be adopted to the new enclosure and test model. We believe this is achievable for laboratories who's emphasis is on employing this approach as a major research line. Envir-onment specific influences can and will degrade performance (such as illumination, occlusions or camera placement). Var-iation in illumination is likely to require additional work and thus our system is unlikely to generalize to, for example, exterior scenes. The dataset we provide, however, can be further augmented to act as a strong starting point for other labora-tories. Another limitation of the present work is that our dataset is based on a limited number ($n = 4$) of macaque subjects. Although these subjects were selected in part to span a range of body morphologies and behavioral phenotypes, it is likely that future studies with other macaques may require additional training, especially if those subjects are atypical in their visual presentation.

It is instructive to compare our system with DeepLabCut[16,17]. The goals of the two projects are quite different—whereas Dee-pLabCut facilitates the development of a tracking system that can track animals, OpenMonkeyStudio is a tracking system. In other words, DeepLabCut provides a structure that helps develop a model; OpenMonkeyStudio is a specific model. DeepLabCut is very general—it can track any of a large number of species; OpenMonkeyStudio only works with macaques. On the other hand, DeepLabCut skirts the major problems associated with monkey tracking—the annotation problem. The solution to these problems constitutes the core innovative aspect of Open-MonkeyStudio. Indeed, the model that results from implementing DeepLabCut will be highly constrained by the training set pro-vided to it; that is, if training examples come from a range of behaviors, it can only track new behaviors within that range. In contrast, OpenMonkeyStudio can track any pose and position a monkey may generate.

Monkeys evolved through natural selection processes to adaptively fit their environments, not to serve as scientific sub-jects. They are, nonetheless, invaluable, and their behavior, in particular, promises major advances, especially if it can be understood in ever more naturalistic contexts[10,34–37]. Despite this, much biological research using them contorts their behavior to our convenience. By reversing things and letting them behave in a more naturalistic and ethologically relevant way, we gain several opportunities. First, we get a more direct correspondence between what we measure and what we want to know—the internal factors that drive the animal's behavior on a moment to moment basis[38]. Second, we gain access to a much higher dimensionality representation of that dataset, which gives us greater sensitivity to effects that are not[39]. Finally, it gives us an ability to measure effects that the monkeys simply cannot convey otherwise. It is for these reasons that improved measure of nat-uralistic behavior holds great hope in the next generation of neuroscience; we anticipate macaques will be part of that step forward.

## Methods

**Training OpenMonkeyStudio**. The main challenge of training a generalizable pose detection model for OpenMonkeyStudio is to collect large-scale annotated data that include diverse poses and viewpoints: for each image, the landmark coordinates of the macaque's pose need to be manually specified. Unlike the pose datasets for human subjects[21–23], this requires primatological knowledge, which precludes collecting a dataset with comparable size (the order of millions; estimated cost: $12 M, assuming a minimum wage of $7.25). One key innovation of Open-MonkeyStudio is the use of the multi-camera system to address the annotation problem through a data augmentation scheme using a theory of multiview geo-metry. The resulting system allows for robust reconstruction of 3D pose even with noisy landmark detections.

**Keyframe selection for maximally informative poses**. Some images are more informative than others. For example, when a macaque is engaged in quiescent repose, its posture will only change modestly over seconds or minutes. After the first image in such a sequence, subsequent ones will provide little to no additional visual information. Including such redundant image instances introduces imbal-ance of the training data, which leads to biased pose estimation. A compact set of the images that include all possible distinctive poses from many views are ideal for the training dataset.

To identify the informative images, we develop a keyframe selection algorithm based on monkey movement, e.g., locomotion, jumping, and hanging. A keyframe is defined here as frame that has large translational movement between its consecutive frames. The translational movement is the 3D distance traveled by the center of mass of a macaque. We approximate the center of mass using the triangulated center of mass. The macaque body is segmented from an image using a background subtraction method that employs a Gaussian mixture model[40], and the center of segmented pixels is computed. The centers of segmented pixels from multiview images are triangulated in 3D using the direct linear transform method[41] given the camera calibration parameters. Robust triangulation using a mean-shift

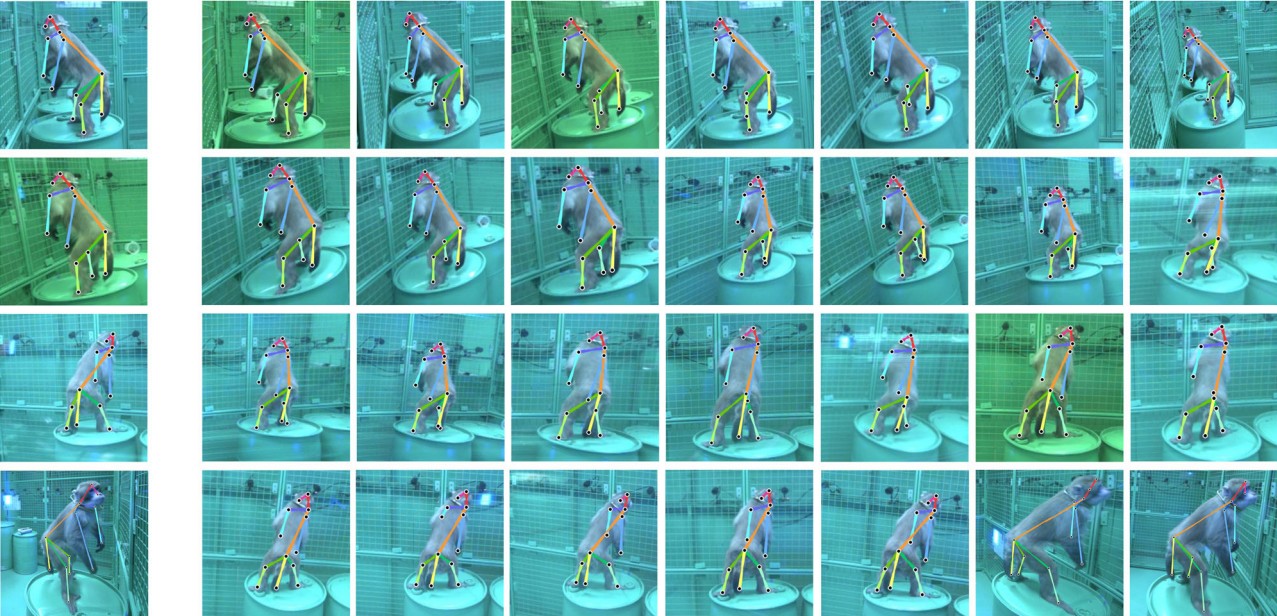

**Fig. 9 Multiview augmentation.** OpenMonkeyStudio leverages multiview geometry to augment the annotated data across views. The three images in the left most column are manually annotated and the 2D pose of the rest images are automatically augmented by 3D reconstruction and its projection.

triangulation approach[42] or random sample consensus (RANSAC[43], see below) can be complementary when background subtraction is highly noisy. With the keyframe selection, the amount of required annotations is reduced by a factor of 100–400, e.g., instead of needing 200,000 labeled frames, we would only need 500–2000.

**Cross-view data augmentation using multiview cameras**. Given the selected keyframes, we annotate the data and extensively augment the size of data using multiview images.

For each keyframe, we crop the region of interest in the images such that the center of mass is located at the center of the cropped region and the window size is inversely proportional to the distance between the center of mass and the camera. By resizing all cropped images to the common resolution (368 width × 368 height), the macaque appears roughly the same size in pixel units. Among 62 view images, we select three to four views that maximize visibility, i.e., most body parts of macaques are visible, and minimize view redundancy, i.e., maximum distance between cameras' optical centers. This selection process is done in a semi-automatic fashion: an algorithm is developed to propose a few camera candidates that the center of mass is visible while retaining the maximal distance between them, and a trained lab member selects the views among the candidates. This selection process significantly alleviates the annotation ambiguity and efforts and reduces the uncertainty of the triangulation by providing a wide baseline between camera optical centers. The set of selected images are manually annotated by the trained annotators. In practice, we leverage a commercial annotation service (Hive AI). As of January 2020, 33,192 images are annotated.

The manual annotations can be still noisy. We use a geometric verification to correct the erroneous annotations. The annotated landmarks are triangulated in 3D using the direct linear transform and projected to the annotated images to check reprojection error, i.e., how the annotations are geometrically consistent. Ideally, the annotated landmarks must agree with the projected landmarks. For the landmark that has reprojection error higher than 10 pixels, we manually adjust the annotations or indicate outliers using an interactive graphical user interface that visualizes the annotated landmarks and their corresponding projections in real time. This interface allows efficient correction of the erroneous annotations across views jointly. The resulting annotations are geometrically consistent even for occluded landmarks. MATLAB code of the adjustment interface is publicly available on our GitHub page.

The refined annotations form a macaque's 3D pose (13 landmarks), which can be projected onto the rest of the views for data augmentation. For example, the annotation of the left shoulder joint in two images can be propagated through any of the other 60 view images collected at that at the same time instant (i.e., same frame count) that include that landmark (i.e., that are not occluded by the body or out of frame). Given our circular arrangement of cameras, this propagation step reduces the amount of annotation needed by a factor of 15–20 depending on the visibility of the 3D landmark location (Fig. 9).

**Training pose detector**. Given the annotated landmarks, we automatically crop the region of the monkey from each image based on the method used in keyframe

selection. When multiple macaques are present, we use a k-means clustering method to identify their centers. We further augment the data by applying a family of nine affine transforms (±30° rotations, 20% left/right/up/down shifting, ±10% scaling, and horizontal flipping) to form the training data. These transformed images enhance the robustness of the detector with respect to affine transformations. The pose detector (CPM) takes as an input a resized color image (368 × 368 × 3 pixel, 368 pixel width and height with RGB channels) and outputs 46 × 46 × 14 pixel response maps (46 width and height with 13 landmarks and one for background). The ground truth response maps are generated by convolving a Gaussian kernel at the landmark location, i.e., in each output response map, the coordinate of the maximum response corresponds to the models' best guess as to the position of the landmark coordinate. $L_2$ loss between the ground truth and inference response maps is minimized to train the CPM. We use ADAM stochastic gradient descent method for the optimization[44]. A key feature of CPM is multi-stage inference, which allows iterative refinements of landmark localization[24]. In particular, such multi-stage inference is highly effective for macaques as the visual appearance of their landmarks (e.g., their hips) is often ambiguous due to the uniform coloration of their pelage. In practice, we use a six stage CPM that produces optimal performance in terms of accuracy and computational complexity. We use a server containing 8 GPU's (NVIDIA RTX 2080 Ti; 11 Gb memory) to train the CPM. Training the model only requires 7 days for 1.1 M iterations with 20 batch size on one card. Hardware on model training (a task usually not repeated often) are quite modest.

**Plausible pose inference**. For the testing (inference) phase, no manual intervention and training is needed. For synchronized multiview image streams of a testing sequence, we compute the 3D center of mass of macaque based on the method used for the keyframe selection and crop the regions of monkeys from multiview images. We localize the landmark position in each cropped image by finding the maximum locations in the response maps predicted by the trained CPM. Given the camera calibration, the landmarks are robustly triangulated in 3D using a RANSAC procedure[43], i.e., for each landmark, a pair of images among 62 images are randomly selected to reconstruct the 3D position that is validated by projecting onto the remaining images. This randomized process allows robustly finding the best 3D position that agrees with the most CPM inferences in the presence of spurious inferences.

The obtained 3D reconstruction is performed on each landmark independently while considering physical plausibility, e.g., limb length must remain approximately constant across time. Given the initialization of the 3D pose reconstruction, we incorporate two physical cues for its refinement without explicit supervision[1]. Limb length cue: for an identical macaque, the distance between landmarks needs to be preserved. We estimate the distance between the connected landmarks (e.g., right shoulder and right elbow) using the median of the distance over time. This estimated distance is used to refine the landmark localization[2]. Temporal smoothness cue: the movement of macaque is temporally smooth over time. The poses between consecutive frames must be similar, which allows us to filter the spurious initialization. We integrate these two cues by minimizing the following

**a**

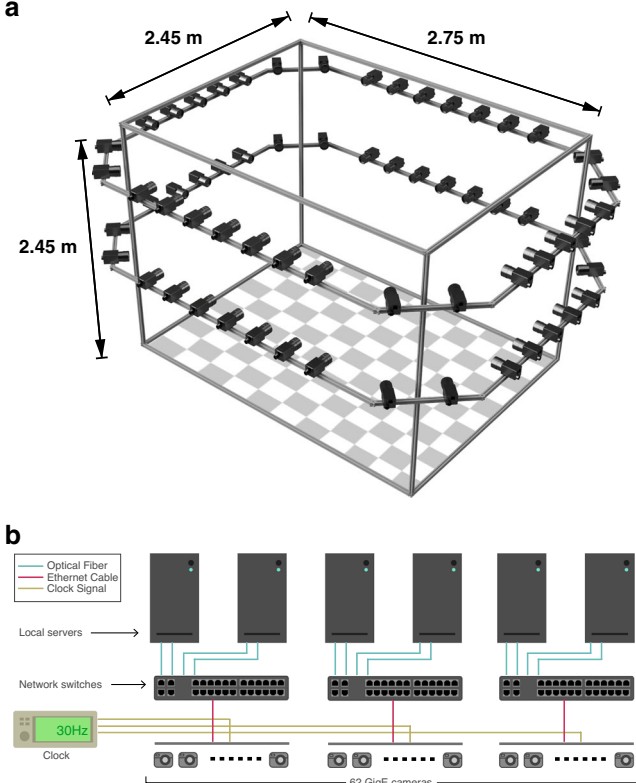

**b**

**Fig. 10 System architecture. a** OpenMonkeyStudio integrates 62 cameras into a large space (2.45 × 2.45 × 2.75 m) that allows unconstrained movement of macaques. These cameras face at the center of space, which is ideal for view augmentation and reconstruction. **b** System configuration of distributed image acquisition. Sixty-two cameras are connected to the local servers through 10 Gb network switches, and six local servers are controlled by a global server. The cameras are triggered by the external clock, which allows synchronization.

objective function:

$$\underset{X_t}{\text{minimize}} \quad \parallel \Pi_i(X_t) - x_{i,t} \parallel^2 + (\parallel X_t - Y_t \parallel - L_{X,Y})^2 + \parallel X_{t-1} - X_t \parallel^2 \quad (1)$$

Where $X_t$ is the 3D location of a landmark at the $t$ time instant, $x_{i,t}$ is the predicted location of the landmark at the $i^{th}$ image, and $\Pi_i$ is the projection operation at the $i^{th}$ image. $Y_t$ is the parent landmark in the kinematic chain, $L_{X,Y}$ is the estimated length between $X$ and $Y$, and $X_{t-1}$ is the 3D location of the landmark at $t-1$ time instant. The first term ensures the projection of the 3D landmark to match with the CPM inference, the second term enforces the limb length constraint, i.e., the distance between adjacent landmarks remain constant, and the third term applies a temporal smoothness. This optimization is recursively applied along a kinematic chain of body, e.g., neck→pelvis→right knee→right foot. For instance, given a root joint (e.g., neck) that is reconstructed without the limb length constraint, its immediate child joints (e.g., pelvis, shoulder, and head) are reconstructed by applying the temporal and limb constraints by minimizing the (equation on Page 9). We use the quasi-Newton method[45] with a threshold value of 1e-05 for successful termination. Then, the reconstructed joints serve as the parent joints to optimize for their child joints (e.g., right knee, elbow, and nose). In practice we first determine an initial estimate (median limb length across one dataset) of the limb lengths for each subject and use the estimate as threshold criteria to find the plausible 3D coordinates for the joints. The kinematic chain is used to apply the thresholds in a structured manner. Over our subject pool, we identified that the neck joint provides a stable 3D estimate and hence was used as the root for the flow structure. Once the root is established, the kinematic chain and the respective thresholds are used to estimate a more accurate 3D joint.

**Multi-camera system design**. Our computational approach is strongly tied to the customized multi-camera system that can collect our training data and to reconstruct 3D pose on the fly (Fig. 10). We integrate the camera system into our 2.45 × 2.45 × 2.75 m open cage at the University of Minnesota. The cameras are mounted on movable arms mounted to a rigid exoskeleton surrounding the cage system and

not touching it (to reduce jitter). The cameras peer into holes in the mesh caging covered with plastic windows. The cameras are carefully positioned so as to provide coverage of the entire system. The resulting system possesses the following desired properties for accurate markerless motion capture: high spatial resolution, continuous views, precise synchronization, and subpixel accurate calibration (mean pixel error = 0.6546, SD = 0.4318).

We use machine vision cameras (BlackFly S, FLIR) that produce a resolution of 1280 × 1024 at up to 80 frames per second (although in practice we use 30 fps). The camera is equipped with a global shutter with sensor size 1/2" format (4.8 μm pixel size). Fisheye lenses with 3–4 mm focal length (Fujinon) are attached to the camera. This optical configuration results in a monkey with 1 m size appearing at ~150 × 150 pixel image patch from the farthest camera (diagonal distance: 5.2 m). Our camera placement guarantees that, in each frame, there exist at least 10 cameras that observe the monkey with a greater than 550 × 550 resolution. This resolution is sufficiently high such that the CPM can recognize the landmark.

Sixty-two cameras are uniformly distributed along the two levels of horizontal perimeter of OpenMonkeyStudio (Fig. 10) made of 80/20 T-slot aluminum (Global Industrial), i.e., for each wall except for the wall with a gate, there are 16 cameras facing at the center of the studio. The baseline between adjacent cameras is ~35 cm, producing less than 6.7° view difference, or 70 pixel disparity at the monkey 3 m away. This dense camera placement results in nearly continuous change of appearance across views where the landmark detector can learn a view-invariant representation, and therefore, can reliably reconstruct landmarks using redundant detection. Further, uniform distribution of cameras minimizes the probability of self-occlusion, e.g., the left shoulder that is occluded by torso for one side of cameras can be visible from the other side of cameras. Positioning of the cameras is performed according to two fundamental principles. First we sample the internal space of the cage with overlapping camera field of views while maintaining a focal length that enables the viewed subject to cover at least half the camera's image sensor. This is done to ensure adequate resolution of the subject. Additionally we configure the cameras in the corners of the rectangular skeleton frame to have a 45° angle. This allows corner cameras to oversample even further which greatly helps with intrinsic and extrinsic camera calibration.

The principle of multiview geometry applies on completely static scenes where the precise synchronization is a key enabler of 3D reconstruction. We use an external synchronization TTL pulse (5 V) that triggers to open and close the shutters at exactly the same moment through General Purpose Input/Output (GPIO). This pulse is generated by a high precision custom waveform generator (Agilent 33120 A) capable of 70 ns rise and fall times. Our system has been extensively tested and remains accurate to sub millisecond precision over 4 h of data acquisition (maximum capacity of our NVMe Raid array)

Geometric camera calibration, the estimate of the parameters of each individual lens and image sensor has to be performed before each recording session. Parameters are used to correct for lens distortions as well as determine the location of the cameras within the scene. To calibrate the cameras, we use a large 3D object (1 m × 3 m) with non-repeating visual patterns (mixed art works and comic strips), which facilitates visual feature matching across views. A standard structure-from-motion algorithm[46] is used to reconstruct the 3D object and 62 camera poses including intrinsic and extrinsic parameters automatically.

**Distributed image acquisition**. Sixty-two cameras produce 3.7 GB data per second (each image is ~2MB with a JPEG lossless compression at 30 Hz). To accommodate such a large data stream, we designed a distributed image acquisition system consisting of six local servers controlled by a global server (Fig. 10). The data streams from 10–11 cameras are routed to a local server (Core i7, Intel Inc.) using individual Cat6 cables to a power over ethernet (PoE) capable network switch (Aruba 2540). The firmware of the switch has been altered by the authors to allow for the specialized requirements of high data throughput using JUMBO packages. Each PoE switch is then connected to the local servers using dedicated fiberoptic 10 Gbit SFP+ transceivers. The data streams are compressed and stored in three solid state drives (NVMe SSD in RAID 0 mode) of the local server.

Cameras also received synchronization pulses through general purpose input output lines (GPIO, Hirose). An individually designed wiring setup provided TTL pulses (5 V) generated at a target frequency of 30 Hz to each camera. Pulses were generated using a high precision custom waveform generator (Agilent 33120 A) capable of 70 ns rise and fall times. Upon completion of a data acquisition session, data are copied onto 12 Terabyte HDD's and physically moved to a JBOD daisy chained SAS hot swappable array (Colfax Storage Solutions) connected to Lambda Blade (Lambda Labs) server.

**Data collection**. All research and animal care was conducted in accordance with University of Minnesota Institutional Animal Care and Use Committee approval and in accord with National Institutes of Health standards for the care and use of non-human primates. Four male rhesus macaques served as subjects for the experiment. All four subjects were fed ad libitum and pair housed within a light and temperature controlled colony room. Subjects were water restricted to 25 mL/kg for initial training, and readily worked to maintain 50 mL/kg throughout experimental testing. Three of the subjects had previously served as subjects on standard neuroeconomic tasks, including a set shifting task[47] and several simple choice tasks[48–52]. Training

also included experience with foraging tasks[53,54], including one study using the large cage apparatus[55]. One subject was naive to all experimental procedures.

Subjects were allowed to move freely within the cage in three dimensions. Five 208 L drum barrels weighted with sand were placed within the cage to serve as perches for the subjects to sit upon. In some sessions, four juice feeders were placed at each of the four corners of the cage in a rotationally symmetric alignment. The juice feeders consisted of a $16 \times 16$ LED screen, a lever, buzzer, a solenoid (Parker Instruments), and were controlled via an Arduino Uno microcontroller. Data were collected in MATLAB via Bluetooth communication with each of the juice feeders. We first introduced subjects to the large cage and allowed them to acclimate to it. Acclimation consisted of placing subjects within the large cage for progressively longer periods of time over the course of about five weeks. To make the cage environment more positive, we provisioned the subjects with copious food rewards (chopped fruit and vegetables) placed throughout the enclosure. This process ensured that subjects were comfortable with the environment. We then trained subjects to use the specially designed juice dispenser[55].

For purposes of comparison with marker data, we collected one large dataset with simultaneous tracking by our OpenMonkeyStudio system and the OptiTrack system. We placed three markers onto a head implant that was surgically attached to the subject's calvarium. This was placed for another study. Briefly, the skin was removed and ceramic screws placed with the bone overlying the crown. A headpost (GrayMatter Research) was placed adjacent to the bone and orthopedic cement (Palacos) was placed around the screws and post in a circular pattern. The marker test took place several years after this procedure. It involved attaching a 3-D printed three-arm holder to the headpost itself. The three arms each bore a reflective marker that could be detected by the Opti-Track system. We used eight Opti-Track cameras (Natural Point, Corvallis, OR) mounted in the same room as our camera system. Placement of the eight cameras was optimized to minimize IR reflections and interference and to obtain a camera calibration (through wanding) error of less that 1 mm.

**Reporting summary**. Further information on research design is available in the Nature Research Reporting Summary linked to this article.

## Data availability

The training dataset is provided and maintained on our github repository (https://github.com/OpenMonkeyStudio).

## Code availability

Our model as well as acquisition and analysis code is provided and maintained on our github repository (https://github.com/OpenMonkeyStudio).

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

## Acknowledgements

We thank Marc Mancarella for critical initial help, Giuliana Loconte and Hannah Lee for ongoing assistance. We also thank Yasamin Jafarian and Jayant Sharma for help with developing the pipelines we used. This work was supported by an award from MNFutures to HSP and BYH, from the Digital Technologies Initiative to H.S.P., J.Z., and B.Y.H., from the Templeton Foundation to B.Y.H., by an R01 from NIDA (DA038615) to B.Y.H., by am NSF CAREER (1846031) to H.S.P., and by a P30 from NIDA (P30DA048742) to B.Y.H. and J.Z.

## Author contributions

B.Y.H., H.S.P., and J.Z. conceived of the research idea. P.C.B., B.R.E., B.Y.H., and J.Z. built the enclosure. B.R.E. trained all animals and collected the dataset. P.C.B., H.S.P., and J.Z. provided all the software code and deep learning. S.B.M.Y. provided essential statistical analysis. P.C.B., B.Y.H., H.S.P., and J.Z. wrote the manuscript and revision.

## Competing interests

The authors declare no competing interests
