## [Peer Review File · Nature Communications]

Reviewers' Comments:

Reviewer #1:

Remarks to the Author:

This article proposes a novel system called OpenMonkeyStudio for markerless 3D pose estimation of freely moving macaques in their cage. The pose estimation is based on a multi-stage Convolutional Pose Machines (10) for the task of articulated pose estimation, which requires a large scale dataset of annotated images for supervised learning. OpenMonkeyStudio has 62 cameras and solves the annotation problem through innovations in image acquisition, annotation and label generation, and augmentation through multiview 3D reconstruction. The performance evaluation was done with OptiTrack, which shows a median reconstruction error of 6.76 cm, and so on. Overall, the system and the experiments were carefully designed, and rigorous analyses were performed for the acquired results. The article is well written. The authors plan to make the developed dataset public, which will be valuable.

Here are the comments.

1. The pose inference procedure should be detailed.

In page 9-10, it is written that "Plausible pose inference" requires the recursive optimization even along a kinematic chain.

Furthermore, there must be the threshold parameter to stop the iterative optimization process.

How to determine the threshold parameter should also be written.

2. The following paper is to be cited, as it is very relevant.

Labuguen, R., et al. Primate Markerless Pose Estimation and Movement Analysis Using DeepLabCut. Proc. 2019 Joint 8th International Conference on Informatics, Electronics & Vision (ICIEV) and 2019 3rd International Conference on Imaging, Vision & Pattern Recognition (icIVPR), pp. 297-300, IEEE, 2019.

3. The authors' interpretation of the proxemics shown in Fig. 7C should be provided.

4. It is written a subpixel accuracy was acquired by the geometric camera calibration. This notion, however, is qualitative. A quantitative measure should be provided.

Reviewer #2:

Remarks to the Author:

The submitted article is about a system and methods for capturing the 3D posture of monkeys without markers using multiple cameras.

This is the first system that captures the motion of monkeys and it is the first system that captures animals in 3D in a systematic way without markers and using multiple view geometry.

It has the potential to boost research in the visual analysis of animal behavior by promoting the multiple view paradigm as a supervising signal for learning.

The authors clearly motivate the approach from the biology experiment perspective: as opposed to humans monkeys cannot wear motion capture suits and markers are not only uncomfortable but are also easily displaced because of the complexity of the monkey body's motion.

A. The main challenge is to learn keypoints at joint positions based only on the RGB appearance. The challenges in such training is that (i) it needs annotated examples (ii) it needs to be invariant to viewpoint and illumination changes as well as to variations between monkey's variation in anatomy and body shape.

Regarding (i) the authors have employed a novel scheme to select the best frames by declaring as

informative poses the starts of significant movement. These annotations are augmented by manual annotation cross views as well as exploiting the epipolar geometric consistency in the frames where the same keypoint is visible (here I have a clarification question, I refer to them as Q1 etc at the end). It is a strength of the paper that at the end the authors/annotations service had to annotated manually a total of 33K images. For (ii) the authors promise a view invariance through the dense sampling using 82 cameras and this is the 2nd strength of multiple views. Regarding other factors it is not clear how the system generalizes cross illumination and animal variation (Q2-3).

The resulted dataset is the first and really unique.

Using these data a well established convolutional pose machine predicts the 2D position of the joints.

B. A geometric inference engine uses the triangulation between cameras with constraints on temporal smoothness and bone length consistency. Here my main question concerns the fact that the approach assumes that the maximum of the heatmap of every keypoint can be chosen as the joint in a deterministic way. Experience shows that many times different joints have the same appearance and the 2D result could be better modelled with a probability density function which might be multimodal (Q4). Authors address this problem by eventually rejecting those through geometric consistency and they are right to do so because they have 82 cameras. However when they reduce the cameras to 8 or 16 this might become a problem.

C. Metric Accuracy: The authors used implanted markers to evaluate the triangulation and at least for the head they had a median error of 6.76cm, which is a great result for a first time monkey experiment and comparable to results on 3D human estimation.

D. It is very worth praising that the authors did a camera "ablation" study by training using a range of viewpoints from 1 to 48 and clearly showed the degradation in accuracy with decreasing number of viewpoints. They also studied the effect of number of viewpoints at the inference triangulation and indeed mention that the arrangement of the 82 cameras might not guarantees as good view invariance as a dome arrangement where the sampling of the viewing sphere is obviously more uniform.

E. Last, the authors used the 3D poses over longer time intervals for behavior classification. By using dimensionality reduction after pose normalization they showed that poses clearly cluster according to behaviors like sitting, standing, and climbing.

F. The system could not be implemented by using Deeplabcut and subsequent triangulation as presented in one of the Deeplabcut papers with triangulated Cheetahs because the use of multiple views in annotation and data augmentation is not supported and is not a trivial extension. The multiple views are the main strength in this article both with respect to creating training examples as well as with respect to the accuracy of 3D triangulation. For this reason the paper is definitely worth publishing.

Authors could clarify the following questions:

Q1: It is not clear how the process called "propagation" works. Given that the system does not segment the silhouette of the monkey or estimates its 3D body shape, it is not clear how visibility cross views can be determined automatically (fig. 9 does not resolve the question). On page 3 they mention that "shoulder and elbow joints indicate hands' occlusion" but they do not elaborate how this happens computationally.

Q2: The authors do not mention any variation in the illumination during training, a severe limitation if the dataset will be used as the basis for generalization in other labs or zoo-like environments.

Q3: The authors used four animals but they do not mention their anatomical or skin variation. A convolutional pose machine learns the local appearance as well as the constellation of the joints that is affected by the particular anatomy. The normalization of the image in pixels would fix the distance scale but not inter-bone length variations.

Q4: Have the authors experienced any situation where the keypoint that was characterized as outlier in their selection scheme was just one of two or three maxima having similar appearance. In general the 3D inference is simple and seems to work but a probabilistic approach might have had even greater accuracy in terms of using soft assignments at the end of the 2D inference.

Kostas Daniilidis

Reviewer #3:

Remarks to the Author:

Summary

This work by Bala et al. describes a new marker-less tracking system for freely moving macaques in large enclosures. It leverages from a sophisticated multi-camera system in combination with the latest algorithms in computer vision to provide a high-resolution 3D location estimate of 13 body landmarks. The system performs as good or even better than commercially available marker-based solutions. The system can be used to infer behaviorally meaningful poses in those animals, and even be extended to pose estimation for multiple animals in parallel, allowing investigation of social behaviors. Overall, this work represents a significant advance in our capacity to quantify behavior in macaques beyond what is currently afforded by ethological tools, and opens the door to neuroscience experiments that could combine this high-frequency behavioral mapping with other high-throughput data streams from neurophysiology.

Major comments

- The main limitation I perceive from this work is its generalizability to other experimental settings. Although the method described by the authors is a major engineering feat, I am skeptical about the ability of other researchers around the world to replicate and adopt the technique described herein. This doesn't diminish in any means the value of the opportunities afforded to the authors by this technology. The public sharing of methods and of the dataset by the authors is commendable. However, after reading the manuscript, I am left to wonder how hard it would be to implement a similar technology in another primate lab. I suggest that the authors, in the Discussion section, mention the potential and associated challenges of replicating this method in another lab. It is not clear, for example, if any part of the algorithm or of the annotated images provided by the authors could be used to train a decoder in a different enclosure with a different background, or if the tools provided are "scene-specific". Discussion of the generalizability of their tool to other settings would be beneficial to appraise to real value of this breakthrough.

Minor Comments

- Page 2. Remove sentence "The thickness and speed of growth of their fur makes skin marking impractical." as it repeats what was said one sentence before.

- Page 3, second paragraph, word missing. "the tracking problem in humans been effectively solved"

- Page 3, third paragraph: "However, while this large number of cameras is critical for training the pose detector, the resulting model can be used in other systems with fewer cameras without training. » It is not clear whether "other systems" refers to other experimental settings (aka. Other scenes), or the same experimental setting with fewer cameras.

- Page 3, paragraph on Accuracy. "Note that there is a spatial bias due to marker attachment (roughly 5 cm above the head) reducing the actual error substantially." This is not clear. Do you mean that the actual error (difference between the two systems) is smaller because of this spatial

bias? Or do you mean that the spatial bias creates an underestimation of the actual error?

- Figure 7B, It is not clear here what should be expected by chance. It doesn't come out clearly from this figure that monkeys correlate their behaviors (that should be represented by a strong diagonal term)

- Figure 7C, Is this data for a single macaque? The figure legends refer to multiple macaques. Also, the maximum of the colorscale does not seem to be represented in the polar plot. Please adjust maximum of scale.

- Page 7, first paragraph, Please comment on the accuracy of multi-individual pose tracking as a function of distance between the two monkeys. Are there more tracking errors when the two monkeys are close to each other?

- Page 7, Discussion, It would be nice to comment on the ability quantify the number of social interactions between the monkeys (chasing grooming, etc.). Can these social behaviors be tracked the same way individual behaviors can?

- Page 7, before-last paragraph, "Although our system is designed for a single cage environment, it can readily be extended to other environment shapes and sizes." This has not been demonstrated in the paper and refers back to my major comment above. It seems that retraining of the algorithm would be required for another environment or scene. Please clarify.

OpenMonkeyStudio - Response to reviewers:

Reviewer #1:

R1.1. The pose inference procedure should be detailed.

In page 9-10, it is written that "Plausible pose inference" requires the recursive optimization even along a kinematic chain. Furthermore, there must be the threshold parameter to stop the iterative optimization process. How to determine the threshold parameter should also be written.

A1.1. We acknowledge that the statement on pose inference was insufficiently detailed and unclear. We have revised it for clarity. Note that we do not use recursive optimization (e.g., dynamic programming) for pose estimation. The following text appears in the revised methods section.

"The optimization is recursively applied along a kinematic chain of the body. Thus, for example it is applied from the neck→pelvis→right knee→right foot. For instance, given a root joint (e.g., neck) that is reconstructed without the limb length constraint, its immediate child joints (e.g., pelvis, shoulder, and head) are reconstructed by applying the temporal and limb constraints by minimizing the (equation on Page 9). We use the quasi-Newton method (Nocedal, J, and S J Wright. 2006) with a threshold value of 1e-05 for successful termination. Then, the reconstructed joints serve as the parent joints to optimize for their child joints (e.g., right knee, elbow, and nose)."

R1.2. The following paper is to be cited, as it is very relevant.

Labuguen, R., et al. Primate Markerless Pose Estimation and Movement Analysis Using DeepLabCut. Proc. 2019 Joint 8th International Conference on Informatics, Electronics & Vision (ICIEV) and 2019 3rd International Conference on Imaging, Vision & Pattern Recognition (icIVPR), pp. 297-300, IEEE, 2019.

A1.2. We apologize for the oversight and thank the reviewer for pointing out this proceeding. It has been incorporated into the paper. Specifically, it now appears in the introduction as:

"For these reasons, the automated measurements of 3D macaque pose is an important goal (13, 14)."

R1.3. The authors' interpretation of the proxemics shown in Fig. 7C should be provided.

A1.3. We apologize for not giving more details about the proxemics analysis, how it was generated and what it shows. We now include the following explanatory text in the revised results section:

"This approach allows us to compute proxemics for two macaques interacting. That is, it lets us compute the average position of each macaque relative to the other, including information about orientation. The relative frequency of a conspecific in a given position relative to a focal subject defines the proxemic relationship of the pair. This information in turn can be used to estimate average distance between subjects, average angle between them, and the interaction of those two terms. Proxemics can be plotted using a radial plot (Figure 7C). This

plot illustrates the proxemics of two macaque subjects. We use the polar histogram of the transformed coordinate to indicate frequency of co-occurrence of the two subjects (color, z-dimension of plot) as a function of their relative distance (r dimension) and angle (theta dimension). “

A thorough analysis and interpretation of the results is beyond the scope of the current manuscript and will require more data as well as detailed replication.

R1.4. It is written a subpixel accuracy was acquired by the geometric camera calibration. This notion, however, is qualitative. A quantitative measure should be provided.

A1.4. Thank you for pointing out this omission. While there is understandably variability in this quantification over the keypoints and data acquisition requires careful calibration for each session, we now provide the metric for all the data sessions that have gone into this manuscript. The numerical result is mean pixel error = 0.6546 (SD = 0.4318).

The manuscript has been updated at the mentioned section to now reads:

“The resulting system possesses the following desired properties for accurate markerless motion capture: high spatial resolution, continuous views, precise synchronization, and subpixel accurate calibration (mean pixel error = 0.6546, SD = 0.4318).”

Reviewer #2:

R2.1: It is not clear how the process called "propagation" works. Given that the system does not segment the silhouette of the monkey or estimates its 3D body shape, it is not clear how visibility cross views can be determined automatically (fig. 9 does not resolve the question). On page 3 they mention that "shoulder and elbow joints indicate hands' occlusion" but they do not elaborate how this happens computationally.

A2.1. As R2 pointed out, we do not explicitly reason about occlusion. Instead, we train the network to predict occluded joints by constructing the occlusion agnostic training data (a similar approach is used for OpenPose). We will add the following figure in the supplementary materials and statement in the manuscript.

"We construct occlusion agnostic training data to handle occlusion, i.e., we use projections of 3D reconstructed pose for labeling data, which enforces the network to predict the locations of occluded joints."

The following figure show the prediction:

The above figure illustrates the PCK curves for occluded and non-occluded joints.

R2.2: The authors do not mention any variation in the illumination during training, a severe limitation if the dataset will be used as the basis for generalization in other labs or zoo-like environments.

A2.2. We acknowledge that the present framework is not designed to generalize to variable illumination, although we do believe that our dataset will be a crucial starting point for other groups interested in solving that problem. We will clarify this in the revised Discussion:

“Environment specific influences can and will degrade performance (such as illumination, occlusions or camera placement). Note that one limitation of the present work is that we did not train our network in a variety of illumination conditions. Variation in illumination is likely to require additional work and thus our system is unlikely to generalize to, for example, exterior scenes. The dataset we provide however can be further augmented to act as a strong starting point for other laboratories.”

R2.3: The authors used four animals but they do not mention their anatomical or skin variation. A convolutional pose machine learns the local appearance as well as the constellation of the joints that is affected by the particular anatomy. The normalization of the image in pixels would fix the distance scale but not inter-bone length variations.

A2.3. We apologize for not providing this information. We used four rhesus macaques selected to span the range of age, hair coloration, and body shape/size, as well as behavioral repertoire. Photographs of the four animals used and the normalized resulting limb length are provided here. The figure is additionally supplied as supplementary material.

Limbs	Kirk	Yoda	Calvin	Timon	Mean	Std_dev
Nose-Head	0.0771	0.0830	0.0947	0.0862	0.0852	0.0073
Head-Neck	0.0960	0.0916	0.1023	0.0986	0.0971	0.0045
Neck-Right_Shoulder	0.0827	0.0761	0.0844	0.0822	0.0813	0.0036
Right_Shoulder-Right_Hand	0.2870	0.2927	0.3854	0.3647	0.3325	0.0500
Neck-Left_Shoulder	0.0827	0.0761	0.0844	0.0822	0.0813	0.0036
Left_Shoulder-Left_Hand	0.2870	0.2927	0.3854	0.3647	0.3325	0.0500
Neck-Hip	0.3609	0.3407	0.4103	0.3814	0.3733	0.0297
Hip-Right_Knee	0.2365	0.2069	0.2749	0.2597	0.2445	0.0296
Right_Knee-Right_Foot	0.1402	0.1439	0.1726	0.1559	0.1532	0.0146
Hip-Left_Knee	0.2365	0.2069	0.2749	0.2597	0.2445	0.0296
Left_Knee-Left_Foot	0.1402	0.1439	0.1726	0.1559	0.1532	0.0146
Hip-Tail	0.2421	0.1860	0.2397	0.1904	0.2145	0.0305

**The measurements are in meters*

As is hopefully also visible from the photographs, the variation in anatomy and fur coloration spans the typical range for laboratory macaques. Thus we anticipate that this dataset will be useful to other scholars. Nonetheless, given the limited supply of macaques at UMN, we did not have the ability to explore a broader range of phenotypes, and this is a limitation of our work. We now acknowledge this limitation in the revised discussion.

“Another limitation of the present work is that our dataset is based on a limited number (n=4) of macaque subjects. Although these subjects were selected in part to span a range of body morphologies and behavioral phenotypes, it is likely that future studies with other macaques may require additional training, especially if those subjects are atypical in their visual presentation.”

R2.4: Have the authors experienced any situation where the keypoint that was characterized as outlier in their selection scheme was just one of two or three maxima having similar appearance. In general the 3D inference is simple and seems to work but a probabilistic approach might have had even greater accuracy in terms of using soft assignments at the end of the 2D inference.

A2.4. Incorporating confidence/uncertainty for triangulation is an excellent idea. We have observed a multimodal distribution of joint prediction heatmap, in particular, when joints are occluded. In this work, we have not incorporated it for the sake of algorithmic simplicity and in practice, it was well handled by RANSAC as there are always a number of views that can observe the occluded joints (as shown in Figure 5). Nonetheless, it is possible to use soft

triangulation (i.e., probabilistic method) to mitigate multimodality or uncertainty of prediction, and it can be used for a future triangulation method that will produce more accurate and stable reconstruction.

Reviewer #3:

R3.1. The main limitation I perceive from this work is its generalizability to other experimental settings. Although the method described by the authors is a major engineering feat, I am skeptical about the ability of other researchers around the world to replicate and adopt the technique described herein. This doesn't diminish in any means the value of the opportunities afforded to the authors by this technology. The public sharing of methods and of the dataset by the authors is commendable. However, after reading the manuscript, I am left to wonder how hard it would be to implement a similar technology in another primate lab. I suggest that the authors, in the Discussion section, mention the potential and associated challenges of replicating this method in another lab. It is not clear, for example, if any part of the algorithm or of the annotated images provided by the authors could be used to train a decoder in a different enclosure with a different background, or if the tools provided are "scene-specific". Discussion of the generalizability of their tool to other settings would be beneficial to appraise to real value of this breakthrough.

A3.1. We thank the reviewer for this thoughtful comment and agree that full generalization is the ultimate goal. In a dream world we have a model running on a mobile device and capture ethologically relevant behavior in the wild. We are far from that future but the steps to get there are already paved. To answer the question in more detail:

We believe that a replication of our enclosure approach is feasible for other labs. It is not yet a trivality of course and a certain degree of engineering effort would be required. Listed are the technical challenges that such a lab would have to undertake:

- 1. Build large scale enclosure and monkey transfer mechanisms.**
- 2. Build networked large scale camera system. (our approach is modular and scalable and can run on personal computers or servers, full system images can be provided).**
- 3. Paint system in homogeneous color.**
- 4. Test calibration routines and arrange cameras such that calibration errors are minimal.**
- 5. Perform initial image acquisition with monkeys.**
- 6. Adopt image segmentation routines to new enclosure and test model.**

We believe this is achievable for laboratories who's emphasis is on employing this approach as their major research line.

We have added the following addition to the discussion section to clarify the reviewers question in greater detail:

“Replication of the system we have developed requires roughly the following technical steps. First a larger scale enclosure (preferably homogeneously painted) and monkey transfer mechanism has to be built. A networked large scale camera system has to be deployed and tested for temporal precision. Calibration routines have to be tested and cameras placed, focused and positioned to minimize calibration errors. Next, initial image acquisitions with monkeys have to be performed. Lastly, image segmentation routines provided in our tools have to be adopted to the new enclosure and test model. We believe this is achievable for laboratories who's emphasis is on employing this approach as a major research line.”

In terms of generalization of the approach: Our tools are certainly not scene specific given the segmenting nature of the approach but occlusions that are fundamentally different or problematic would change that. Providing both the model, the code and the dataset hopefully

aids this effort. Dr. Mackenzie Matthis and the DeepLabCut team have recently released a model zoo approach to generalization and while it is too early to tell how well this approach will work, we are hopeful that this will lead to very general models in the future. Currently we are exploring teacher-student (such as Hinton et al. 2015, <https://arxiv.org/pdf/1503.02531.pdf>) learning methods that can aid in that process (e.g. what has to be relearned between environments if anything).

We have added a more in depth discussion of these thoughts in the discussion section of the manuscript that reads:

“OpenMonkeyStudio is designed as a blueprint for tracking poses in monkeys. As such, engineering expertise is essential in replicating our approach. Our analysis provides strong evidence that generalization to other laboratory environments is possible, however we are currently not providing a plug and play solution. Environment specific influences can and will degrade performance (such as illumination, occlusions or camera placement). The dataset we provide however can be further augmented to act as a strong starting point for other laboratories.”

Minor Comments

- Page 2. Remove sentence “The thickness and speed of growth of their fur makes skin marking impractical.” as it repeats what was said one sentence before.

R. Thank you, the redundancy has been fixed.

- Page 3, second paragraph, word missing. “the tracking problem in humans been effectively solved”

R. Thank you, the sentence now reads “the tracking problem in humans has been effectively solved”

- Page 3, third paragraph: “However, while this large number of cameras is critical for training the pose detector, the resulting model can be used in other systems with fewer cameras without training. » It is not clear whether “other systems” refers to other experimental settings (aka. Other scenes), or the same experimental setting with fewer cameras.

R. Thank you for pointing out this lack of clarity. We have updated the text to read:

“However, while this large number of cameras is critical for training the pose detector, the resulting model can be used in other systems (for example by other laboratories) with fewer cameras without training.”

- Page 3, paragraph on Accuracy. “Note that there is a spatial bias due to marker attachment (roughly 5 cm above the head) reducing the actual error substantially.” This is not clear. Do you mean that the actual error (difference between the two systems) is smaller because of this spatial bias? Or do you mean that the spatial bias creates an underestimation of the actual error?

R. We apologize for being unclear. The reviewer is correct in assuming that the actual error should be substantially lower because of this offset. Furthermore, our training

dataset did not include any instrumented animals which could further amplify errors in this analysis. We have reformulated the sentence and added additional clarification:

“Note that there is a spatial bias due to marker attachment (roughly 5 cm above the head) which inflates the presented error estimate. In addition to this systematic offset, our training data did not include instrumented (implant carrying) animals which could further increase variability in the location estimate.”

To make the displacement more apparent to the reviewer we share a close up frame of the animal wearing the marker on their head implant. The figure is also supplied as supplementary material.

Figure for reviewer. Close up of one camera showing one rhesus macaque wearing a reflective marker implant above the head.

- Figure 7B, It is not clear here what should be expected by chance. It doesn't come out clearly from this figure that monkeys correlate their behaviors (that should be represented by a strong diagonal term)

R. We thank the reviewer for pointing this out. We meant that expression as a qualitative statement. We do not compute a null distribution nor do we make a statistical assessment of the result. Here we demonstrate the type of analysis that can be performed on these datasets. In the revision, we have removed the terminology and now refer to: **Co-occurrence of actions in social macaques.**

- Figure 7C, Is this data for a single macaque? The figure legends refer to multiple macaques. Also, the maximum of the colorscale does not seem to be represented in the polar plot. Please adjust maximum of scale.

R. The data refers to one animals' proximal position to the other. We have now included additional explanatory text in the revised Results to explain what this analysis is.

“This approach allows us to compute proxemics for two macaques interacting. That is, it lets us compute the average position of each macaque relative to the other, including information about orientation. The relative frequency of a conspecific in a given position relative to a focal subject defines the proxemic relationship of the pair. This information in turn can be used to estimate average distance between subjects, average angle between them, and the interaction of those two terms. Proxemics can be plotted using a radial plot (Figure 7C). This plot illustrates the proxemics of two macaque subjects. We use the polar histogram of the transformed coordinate to indicate frequency of co-occurrence of the two subjects (color, z-dimension of plot) as a function of their relative distance (r dimension) and angle (theta dimension).”

We thank the reviewer for pointing out the mistake on the colorbar. It has been updated accordingly.

- Page 7, first paragraph, Please comment on the accuracy of multi-individual pose tracking as a function of distance between the two monkeys. Are there more tracking errors when the two monkeys are close to each other?

R. This is an excellent question. For the time being the answer is no. Given the method we currently employ (segmentation of the individuals) there are no additional errors as long as the segmentation works. If we had animals (comment below) that engaged in complex social interactions such as grooming, tool sharing, rearing, we do anticipate a significantly larger error.

- Page 7, Discussion, It would be nice to comment on the ability to quantify the number of social interactions between the monkeys (chasing grooming, etc.). Can these social behaviors be tracked the same way individual behaviors can?

R. This is an excellent comment and it is the subject of ongoing research by our lab. For several reasons that may not be obvious, doing so requires some innovations that are beyond the scope of the present work. However, from a technical standpoint the problem is solved - the data are good enough for this purpose, even if the analysis is not. We anticipate it may take two years of work by a smart post-doc to solve these problems.

- Page 7, before-last paragraph, "Although our system is designed for a single cage environment, it can readily be extended to other environment shapes and sizes." This has not been demonstrated in the paper and refers back to my major comment above. It seems that retraining of the algorithm would be required for another environment or scene. Please clarify.

R. We thank the reviewer for this comment and share the reviewer's concern. As long as it is possible for other labs to obtain images at comparable image resolutions as used in our network, and given other laboratories abilities to adjust the image segmentation (cropping) parameters to their environment, we do not believe there to be any need for retraining.

Reviewers' Comments:

Reviewer #1:

Remarks to the Author:

I appreciate the author's efforts for this revision.
All the responses seem to be sufficient.

Reviewer #2:

None

Reviewer #3:

Remarks to the Author:

The authors have answered all my questions. Great work!

OpenMonkeyStudio - Response to reviewers:

R1.1. I appreciate the author's efforts for this revision. All the responses seem to be sufficient

A1.1. We thank the reviewer for their time and careful consideration of our manuscript.

R2.1. The authors have answered all my questions. Great work!

A2.1. We thank the reviewer for their time and complements.